# Exploring the Effect of Ammonium Iodide Salts Employed in Multication Perovskite Solar Cells with a Carbon Electrode

**DOI:** 10.3390/molecules26195737

**Published:** 2021-09-22

**Authors:** Maria Bidikoudi, Carmen Simal, Vasillios Dracopoulos, Elias Stathatos

**Affiliations:** 1Department of Electrical and Computer Engineering, University of the Peloponnese, 26334 Patras, Greece; mbidikoudi@uop.gr (M.B.); c.simal@uop.gr (C.S.); 2Foundation for Research & Technology, HELLAS (FORTH), Institute of Chemical Engineering Sciences (ICE-HT), Platani Rio, 26504 Patras, Greece; indy@iceht.forth.gr

**Keywords:** ammonium salts, perovskites, carbon electrode, perovskite additives, post treatment, recombination, perovskite solar cells

## Abstract

Perovskite solar cells that use carbon (C) as a replacement of the typical metal electrodes, which are most commonly employed, have received growing interest over the past years, owing to their low cost, ease of fabrication and high stability under ambient conditions. Even though Power Conversion Efficiencies (PCEs) have increased over the years, there is still room for improvement, in order to compete with metal-based devices, which exceed 25% efficiency. With the scope of increasing the PCE of Carbon based Perovskite Solar Cells (C-PSCs), in this work we have employed a series of ammonium iodides (ammonium iodide, ethylammonium iodide, tetrabutyl ammonium iodide, phenethylammonium iodide and 5-ammonium valeric acid iodide) as additives in the multiple cation-mixed halide perovskite precursor solution. This has led to a significant increase in the PCE of the corresponding devices, by having a positive impact on the photocurrent values obtained, which exhibited an increase exceeding 20%, from 19.8 mA/cm^2^, for the reference perovskite, to 24 mA/cm^2^, for the additive-based perovskite. At the same time, the ammonium iodide salts were used in a post-treatment method. By passivating the defects, which provide charge recombination centers, an improved performance of the C-PSCs has been achieved, with enhanced FF values reaching 59%, which is a promising result for C-PSCs, and V_oc_ values up to 850 mV. By combining the results of these parallel investigations, C-PSCs of the triple mesoscopic structure with a PCE exceeding 10% have been achieved, while the in-depth investigation of the effects of ammonium iodides in this PSC structure provide a fruitful insight towards the optimum exploitation of interface and bulk engineering, for high efficiency and stable C-PSCs, with a structure that is favorable for large area applications.

## 1. Introduction

Perovskite Solar Cells (PSCs) have been in the spotlight of research over the past 10 years, which has led to significant improvements in the technology’s output. In particular, PSCs were first reported in 2009, as an evolution of the Dye Sensitized Solar Cell (DSSC) technology. The newly introduced solar cells achieved Power Conversion Efficiency (PCE) of 3.9% [1] and after intensive experimental work on the perovskite structure, the device configuration and fundamental studies on the working principles and physical properties of these materials [2,3,4,5,6,7], PCEs of 25.6% have now been recorded in monolithic devices [8].The factors that contribute to the high efficiency of perovskites, as active materials for solar cells, are their unique optoelectronic properties, including enhanced light absorption, high charge carrier mobility and long carrier lifetimes [9,10]. Even though huge progress has been made, PSCs still lag behind their theoretical PCE, determined by the Shockley-Queisser limit, which is about 31% [11].

The main reason for this power conversion loss is the recombination processes that take place during the operation of PSCs, which affect negatively the open circuit voltage (V_oc_) and the fill factor (FF) obtained, reducing correspondingly the overall device efficiency [12]. In particular, non-radiative recombination, also known as Shockley–Read–Hall (SRH), that accounts for the largest portion of losses, occurs both at the interfaces between the perovskite and the charge transport layers, as well as the perovskite bulk and the grain boundaries [13,14]. The most promising approaches that have been used in order to minimize the trap-assisted non-radiative recombination losses are the control of the perovskite crystallization, through the engineering of the perovskite composition and the use of additives, the defect passivation and interface engineering and the formation of graded 2D/3D junctions, by applying post-treatment methods [15,16,17]. By applying the aforementioned techniques, increases in the PCE have been achieved, reducing the gap between the theoretical and realistic PCEs obtained.

Organic molecules with several functional groups, the most prominent being ammonium iodide salts [18], have been used to successfully passivate defects. Alkyl- and arylammonium halides have been effectively incorporated as additives into the perovskite precursor solution and have been proven to be beneficial for the film formation, resulting in improved surface morphology and enhanced crystallinity of the active phase. These organic amine salts (R^+^NH_3_ X^−^), which possess electron-deficient H^+^ with the ability to accept electrons, interact with the perovskite film surface via hydrogen bonding between the ^+^NH_4_ group and the highly nucleophilic [PbI_6_]^4−^ octahedra of the halide-based perovskite structure. The strong binding between the organic and inorganic frameworks is believed to regulate the grain size, consequently affecting the film coverage and crystal growth in the mesoporous scaffolds. Furthermore, it is well known that the counter ion on these organic ammonium halide modulators, usually I^−^, can passivate halide defects, thus mitigating the potential recombination of the charge carrier at the interface between the perovskite absorber and the load carrying layers [19].

Among these, phenethylammonium iodide (PEAI) and tetrabutyl ammonium iodide (TBAI) have emerged as highly efficient, achieving PCEs of 23.3%, with Voc of 1.18 V [20] and 21.6 mA/cm^2^ with Voc of 1.12 V respectively [21,22], which is over 90% of the Shockley-Queisser Voc limit. The working mechanism and the role of ammonium iodides, both as additives and as passivation agents, has been widely studied in typical PSCs [18,19,23,24,25,26,27].

By introducing the ammonium iodides as additives in the perovskite precursor solution, chemical interactions between the additive and the perovskite occur, with the most prominent being the hydrogen bonding between the inorganic framework and the ammonium group. This interaction is responsible for the ordering and packing of the molecular domains, while it additionally determines the geometry and properties of the perovskite crystal, by regulating the crystal growth [28,29].

On the other hand, the post-treatment of the perovskite film, in the annealed perovskite phase, using the ammonium iodides as post-treatment agents is expected to reduce the trap density at the surface of the perovskite layer and consequently improve the performance and stability [16,30]. By using this method, a passivation occurs at the perovskite surface, which makes the material less prone to external environment effects and a suppress of the ion movement, together with reduction of defects, is achieved [31].

Herein, a series of ammonium iodide salts have been implemented in PSCs, which use carbon (C) as the counter electrode and are referred to as Carbon-based Perovskite Solar Cells (C-PSCs). This type of device differs from the metal electrode-based PSCs, since there is a complete elimination of the typical gold (Au) and/or silver (Ag) metal electrode, which is substituted by a C electrode, prepared from a C paste. In addition, owing to the bipolar charge carrier transport properties of perovskites, the expensive and unstable hole transport layer (HTL) can be omitted from these devices. C-PSCs offer the advantages of low cost and scalability, combined with ambient processability and increased device stability, which make them highly promising for commercialization.

In particular, in this work, the ammonium salts of ammonium iodide (AI), ethylammonium iodide (EAI), tetra butylammonium iodide (TBAI), phenethylammonium iodide (PEAI) and 5-ammonium valeric acid iodide (AVAI) have been used as additives in the multiple cation-mixed halide perovskite precursor solution, in order to control the crystallization process. Moreover, the same organic compounds have been used to post-treat the perovskite film, with the scope of serving as passivating agents. Both their use as additives and as passivators has been applied and evaluated in C-PSCs of the triple mesoscopic structure. A study of the ammonium iodide addition effects has been performed, together with a comparison among the different anions used. Being targeted in the triple mesoscopic C electrode-based PSC architecture, which is the most promising for large area applications but also the most challenging in terms of the fabrication methods and processing, this study provides a useful insight into the different routes towards the increase of the PCE and at the same time highlights the peculiarity and uniqueness of this system compared to the conventional metal electrode PSCs.

## 2. Results and Discussion

### 2.1. The Effect of Ammonium Iodides as Additives

#### 2.1.1. X-ray Diffraction (XRD) Analysis

In order to evaluate the effect that the addition of ammonium iodides in the perovskite precursor solution has on the resulting perovskite crystals’ properties, X-ray Diffraction (XRD) measurements have been performed in full stack devices of the triple mesoscopic structure (c-TiO_2_/m-TiO_2_/ZrO_2_/perovskite/C) and the results have been analyzed (Figure 1).

All the perovskite films, with or without the additives, exhibit the perovskite cubic phase peaks at ~14.3°, 28.6° and 32°, corresponding to the (100), (200) and (210) planes respectively. The intense peak at 2θ = 26.8° is assigned to the graphite present in the C electrode [32,33,34], while the peaks at ~24.7° and ~38° originate from the anatase phase of TiO_2._ Finally, the peak at ~35.1° originates from the ZrO_2_.

A small peak of unreacted PbI_2_ is observed at ~12.8° at the spectra of the reference perovskite, which disappears after the insertion of additives, indicating their beneficial effect on suppressing the formation of inactive species, and confirming the interaction of the ammonium cation with the iodine defects that are present during the perovskite formation. An increase in the intensity of the perovskite peaks is noted for all additives, except the AI, which is indicative of the expected increase in the photocurrent obtained for the corresponding C-PSCs. A 2D perovskite component appears at 7.8° for the TBAI additive [35]. It is contemplated that the TBA cation could be pushed vertically towards the surface of the perovskite grains due to its large size and high level of hydrophobicity compared to the other additives under study, being therefore more likely to form a 2D layered phase in the perovskite/C interface. However, this peak is weak and is not present in the rest of the ammonium iodides under study. This indicates that the 2D perovskite is not formed at these conditions and any improvement in the C-PSCs performance should be attributed to the improved crystal quality rather than the composition of the perovskite, which in all cases appears to be of the 3D structure.

#### 2.1.2. Ultraviolet–Visible Spectroscopy

To investigate the optical properties of the as-prepared perovskite thin films, and to calculate the bandgap (Eg) in each case, we conducted ultraviolet-visible (UV-vis) absorption spectroscopy measurements, as displayed in Figure 2a. In order to determine the bandgap, the Tauc method was used, where (αhν)^2^ is plotted as a function of energy hν (Figure 2b) and the linear region is fitted, so that the bandgap results from an extrapolation of this linear fit to the x-axis [36,37].

All the additive-containing perovskite films exhibit a stronger absorption than the reference perovskite film, indicating an enhanced light-harvesting ability of the corresponding perovskites and implying a better performance, with an increase in the generated photocurrent density values when incorporated in C-PSCs, while the highest absorbance is noted for the AI-based perovskite film. The increased absorbance of the films is an indication of high quality perovskite crystals, with increased coverage and uniformity [38,39]. The absorption band edge is nearly identical for all perovskites under study and similar values of Eg have been calculated, demonstrating that the addition of ammonium iodides does not affect the energy levels of the perovskite.

#### 2.1.3. Scanning Electron Microscopy (SEM)

In order to study the effect of the additives on the surface morphology, Scanning Electron Microscopy (SEM) measurements were performed on half-cells of the FTO/c-TiO_2_/m-TiO_2_/ZrO_2_/perovskite structure.

This structure has been chosen in order to clearly elucidate the effect of additives on the reorganization and crystallization of the perovskite only, without the additional effect of the C electrode. When observing the SEM pictures obtained for the perovskite films prepared from the precursor solution with the ammonium iodide additives under study (Figure 3), no significant morphological changes are noticed compared to the films obtained from the reference perovskite precursor solution. All films appear dense, with few pinholes and a similar crystal size. This leads to the conclusion that any changes in the corresponding C-PSCs performance should be attributed to the changes in the physical and chemical interaction of the precursor solutions with the C electrode and the crystallization process that occurs after the infiltration of the perovskite precursor through it. This result highlights the complexity of the triple mesoscopic C-PSC configuration and justifies the intensive research that needs to be performed, which is in a completely different route compared to metal electrode-based PSCs.

#### 2.1.4. Solar Cells’ Characterization

The 6 different perovskites have been incorporated in C-PSCs of the triple mesoscopic structure (c-TiO_2_/m-TiO_2_/ZrO_2_/perovskite/C) and the typical photocurrent-voltage (J-V) curves are recorded and presented in Figure 4, while the corresponding photovoltaic parameters are summarized in Table 1.

The varied cations (A^+^, EA^+^, TBA^+^, PEA^+^ and AVA^+^) intercalate between MA^+^/FA^+^ and [PbI_6_]^4−^ through diverse intermolecular interactions (mainly hydrogen bonds), affecting the quality of perovskite films. The strength of this intermolecular hydrogen bonding is expected to have an effect on the crystallization kinetics, slowing down the crystal growth rate during the thermal annealing, and delivering a more ordered and homogeneous perovskite film, with fewer unordered aggregations of perovskite particles on the mesoporous films [19,40].

A significant increase in the photocurrent density values obtained is noted, when AI is used as an additive to the reference perovskite, reaching values of 23.9 mA/cm^2^, much higher than 19.8 mA/cm^2^, which has been obtained with the reference perovskite. Combined with a 20 mV increase of the Voc value and FF of 0.51 vs. 0.47 for the reference device, the AI-based PSC has achieved the highest PCE that has been recorded in the frames of this study, reaching 10.3%. This is in accordance with the highest absorption values that the AI-based films have exhibited and is attributed to the ability of the small sized ammonium cation to neutralize the negatively charged iodine defects of the halide-based perovskite structure via electrostatic interactions, involving ionic bonding and hydrogen bonding (H_3_N–H···I–PbI_5_) [41,42]. High photocurrent value of 24 mA/cm^2^ has been recorded for the AVAI-based perovskite; however, a simultaneous drop in both the V_oc_ and FF values has not resulted in increased PCE of the particular devices. A similar trend has also been observed for the ammonium iodide, ethylammonium iodide, tetrabutylammonium iodide and phenethylammonium iodide additives: an increase in the photocurrent density values with a decrease in the FF values and a simultaneous decrease in the V_oc_ value for the PEAI additive, which sacrifices the PCE obtained and is restrained lower than the reference perovskite values of 7.4% for EAI, 7% for TBAI and 5.73% for the PEAI respectively. The varied impact of the studied organic amine salts on the cell efficiency of the respective device is justified by the different sizes of the (alkyl/aryl)ammonium ion (R-^+^NR’_3_) used in each case. The smaller size and higher symmetry that NH_4_I possesses compared to the other organic ammonium iodides under study facilitates the approach of the ammonium cation, ^+^NH_4_, to the perovskite octahedra surface, thus favoring their interaction. As a result of this strong interaction, the crystal growth during the annealing process is assisted, while additionally, NH_4_I is expected to volatilize easily during thermal annealing, thus avoiding any undesirable excess additives that may lead to poor performance. Moreover, the small size of AI does not affect the proper infiltration of the perovskite precursor solution through the mesoporous C layer, thus providing a complete filling of the pores and homogeneous coverage. When large-sized cations are employed, including TBA^+^ (R = R’ = n−Bu) and PEA^+^ (R = PhCH_2_CH_2_, R’ = H), the presence of such large alkyl/aryl groups resulted to be detrimental for the C-PSCs performance, due to the minimum penetration through the triple mesoscopic stack. The large hydrophobic alkyl chains prevent the perovskite precursor solution from penetrating the three mesoporous layers (C, ZrO_2_ and TiO_2_) causing the perovskite crystal to grow superficially during the thermal treatment. This could account for the much-decreased observed FF values and lower efficiencies compared to the reference device. Finally, it should be mentioned that the reduced photovoltaic performance of PEAI-based devices compared to those of the TBAI, which is also the lowest PCE obtained, could be attributed to the lower sublimation temperature of PEAI with regards to TBAI, which causes the organic spacer to volatilize faster from the perovskite film, thus having a smaller impact on the nucleation and crystal growth [43].

The J-V curves of each device, in the forward and reverse scanning mode, are presented in Figure 5 and the hysteresis index (HI) of the devices with and without the ammonium iodide additives have been calculated by the formula $Hysteresis\;Index=\frac{PCE_{RS}-PCE_{FS}}{PCE_{RS}}$ (Table 2). A significant improvement in the HI has been noted for all the ammonium iodides. This effect has been previously reported in the literature and has been attributed to the phase stabilization of the perovskite formulations and an effective suppression of charge recombination [18,41]. In addition, the NH_4_^+^ cation has been proposed to react with [PbI_6_]_4_, reducing the migration of ions in perovskite and thus improving the hysteresis [24]. Even though the lowest HI has been noted for the TBAI, this result is not reliable, since it is a low-performing device with an FF that is already not appropriate for high efficiency C-PSCs of the triple mesoscopic structure, while it has been obvious even by optical observation that this perovskite has limited penetration through the triple mesoscopic stack.

#### 2.1.5. Electrical Characterization

In order to shed light onto the changes that the ammonium additives provoke on the electrical properties of the as prepared C-PSCs, the Electrochemical Impedance Spectroscopy (EIS) spectra of the C-PSCs have been recorded, under illumination at an applied voltage of 0.8 V (close to the V_oc_). The Nyquist plots obtained are illustrated in Figure 6a and the Bode phase pots are illustrated in Figure 6b. The results obtained after the fitting of the spectra, by using an equivalent electrical circuit (Figure 7), are summarized in Table 3. The Bode phase plot is a useful depiction of the number of equivalent electrical elements that the circuit comprises of, while the Nyquist plot provides information on the type of these elements.

The equivalent electrical circuit that has been used to fit the data obtained from the EIS measurements composes of a series resistance, R_s_, an R-C circuit, related to the charge transfer at the contacts (C electrode, ETL/perovskite and perovskite/C interface), denoted as R_ct_, and the impedance due to bulk perovskite, Z_G_, depicted by a Gerischer element in order to obtain a better fit.

In all C-PSCs under study the Nyquist plots, imaginary vs. real part of impedance (Z′ vs. Z′′) present two distinctive semicircles, while the Bode phase plots present, accordingly, two peaks. The series resistance (R_s_) correlates with the resistance from the contacts and is depicted as the first point of interception of the spectra with the Z’ axis. A low Rs value, compared to the reference, is noted for the AI perovskite, in accordance with the higher FF values that these devices exhibit, while the highest Rs value has been recorded for the PEAI perovskite, which has exhibited the lowest FF value when incorporated in C-PSCs.

The R_ct_ resistance, obtained from the fitting of experimental data of the high frequency (HF) arc in the Nyquist plot, is related to the charge transfer resistance between the perovskite and selective contacts, i.e., charge transport in the mesoporous TiO_2_, ZrO_2_ and C electrode layer, as well as at the perovskite/ETL and perovskite/C interfaces [44,45,46]. The two perovskites that exhibit the highest J_sc_ values when incorporated in C-PSCs, which are AI-based and the AVAI-based, are also present in the lowest R_ct_, which justifies an optimized charge transport in these cases. On the contrary, the highest R_ct_ value has been recorded for the TBAI perovskite, which has proven to be not suitable for C-PSCs of the triple mesoscopic structure, due to the inability of penetrating through the triple mesoscopic stack.

At low frequencies, a Gerischer pattern is identified, which appears as a straight line with a 45° slope [47] followed by an arc and is mainly present in devices where the diffusion length is shorter than the sample thickness. On the Nyquist-plot, it resembles a thin-layer diffusion element and is given by the equation: $Z_{G}\left(\Omega \right)=\frac{1}{Y_{0}\sqrt{k_{a}+j\omega }}$, where Y_0_ is the admittance parameter and k_a_ is the reaction rate constant. The Gerischer impedance Z_G_ is inversely proportional to the recombination rate [48] and provides information about the recombination processes. Even though the highest Z_G_ value has been recorded for the PEAI based perovskite, this is not reflected in the performance of the C-PSCs, which highlights the great importance of the appropriate materials’ selection, when preparing C-PSCs of this particular structure. Even though the electrical parameters of PEAI seem favorable for application in such devices, the chemical limitations for the proper incorporation in the triple mesoscopic stack have yielded C-PSCs, which are the lowest performing among the systems under study. On the contrary, the lowest Z_G_ value has been recorded for the highest performing additive, AI; however, this simultaneous increased the ability for better charge extraction compensates in this system, providing the highest performing device.

In the Bode phase plots, the 2 peaks that are located in the low-frequency region and the high-frequency region correspond to the equivalent semicircles noted at the Nyquist plots in the same frequency regions, the recombination process at the low frequency and the charge transfer process at the high frequency parts. There is an obvious increase in the peaks of the spectra obtained for TBAI- and PEAI-based perovskites, indicating an increased capacitive behaviour for these devices, while the shifts of the frequency where the peaks are located correlates with the electron lifetime, by the formula $\tau =\frac{1}{2\pi f}$, further explaining the low performance of PEAI-based perovskite, which has the lowest τ [49,50].

Overall, the EIS study of the additive-based C-PSCs has highlighted the significance of physical limitations and the challenges that need to be taken into account and studied in detail for the successful fabrication of this type of device, considering that the chemical interactions of perovskites with the mesoscopic layers, and especially the mesoporous carbon electrode, can have a detrimental effect on the corresponding PSC performance, in contrast to the conventionally structured metal electrode PSCs.

#### 2.1.6. Stability

In order to investigate the effect of the additives in the stability of the as-prepared C-PSCs, the highest performing devices have been stored in a dark shelf, under ambient conditions (temperature ~35 °C and humidity ~40%) for 60 days and the J-V measurements were recorded (Figure 8) and the electrical parameters have been obtained (Table 4).

One of the most attractive and promising features of C-PSCs is their remarkable stability under ambient conditions, a feature that has also been confirmed in this study as well. As can be seen from the electrical parameters extracted from the J-V curves of the C-PSCs with and without the use of ammonium iodide additives, the performance of the devices presents small diminishment after 60 days of shelf life. In particular, the highest PCE loss has been of 26%, which classifies the device as stable, considering that the whole fabrication process and storage has been performed entirely under ambient conditions and considering the degradation that the typical metal cathode-based PSCs present, which is only a few hours after their preparation. The reference device, as well as the AVAI-based device, present a slight increase in the PCE values, mainly driven by the increase of the photovoltage values, owing to the self-induced recrystallization and healing that has been previously reported [51,52,53]. However, an interesting result, which is in contrast to the literature reports on the stability of ammonium-based perovskites in PSCs so far, is that the addition of ammonium iodides does not enhance the stability of the devices. The additive-based perovskite C-PSCs present a drop in the PCE after 60 days of storage, with the cause of this drop being the decrease of the photocurrent density values. An exception to the drop in the PCE is the AVAI additive, where the combined increase in the Voc and FF values compensate for the photocurrent degradation, resulting in a small increase in the PCE. The drop in the photocurrent values with time can be attributed to the excess of ammonium cations present in the additive-based perovskites, which are prone to multiple degradation reaction routes, as proposed by Juarez-Perez et.al. [54] Overall, ammonium iodides as additives do not have a positive effect on the stability of the C-PSCS, even though the devices can be classified as stable.

### 2.2. Ammonium Iodides as Post-Treatment Agents

#### 2.2.1. X-ray Diffraction (XRD) Analysis

X-ray Diffraction (XRD) measurements have been performed in full stack devices of the triple mesoscopic structure (c-TiO_2_/m-TiO_2_/ZrO_2_/perovskite/C), in order to investigate the effect that the applied post-treatments have on the structural properties of the resulting perovskite films and the results have been analyzed (Figure 9).

All the perovskite films exhibit the 3D perovskite cubic phase peaks at ~14.3°, 28.6° and 32° and the intense graphite peak at 2θ = 26.8°, while the peaks at ~24.7°, ~38° and ~35.1° originate from the anatase phase of TiO_2_ and the ZrO_2_. A small peak of photoinactive δ phase caused by unreacted PbI_2_ is found at ~12.8° for the reference perovskite film, which is diminished after the post treatment with ammonium iodides and completely disappears in the AVAI post-treated sample. This is justified by the well-known ability of AVAI to improve the moisture stability of PSCs, which is crucial when preparing perovskite devices under ambient conditions [55].

A notable peak, which is assigned to the formation of 2D perovskite, is noted at 7.6° after the TBAI post-treatment, implying that the 2D perovskite phase (TBA)PbX_3_ coexists with the 3D perovskite phase, which is consistent with the results previously reported in the literature [56]. However, the performance of the corresponding C-PSCs directly depends on the infiltration ability of the perovskite precursor solution through the triple mesoscopic stack, and even though it appears that only TBAI is capable of forming a 2D capping layer, the chemical and physical interactions that affect the perovskite/charge transport interfaces are the factors that determine the devices’ performance.

#### 2.2.2. Ultraviolet–Visible Spectroscopy

In order to investigate the impact of the post-treatment on the optical properties of the as-prepared perovskite films, and to evaluate the potential changes in the energy bandgap (Eg) in each case, we performed ultraviolet-visible (UV-vis) absorption spectroscopy measurements, at half cells of the c-TiO_2_/m-TiO_2_/ZrO_2_ structure, as displayed in Figure 10a. The bandgap has been determined by using the Tauc method, as described in Section 2.1.2 (Figure 10b).

As can be seen in Figure 10b, the post-treatment methods do not affect the bandgap of the perovskite, where minor changes have been observed with regards to 1.578 eV that has been calculated for the reference perovskite. This has been an expected outcome, which agrees with the literature [18] and justifies that the post-treatment method mainly affects the surface morphology and properties, without altering the bulk of the perovskite. In Figure 10a, a distinctive increase in the absorption has been noted for the TBAI and PEAI compounds. This is a result of the improved perovskite film morphology, with passivated defects, both originating from the grain boundaries (2D) and iodine and lead clusters (3D) [15] and a suggestion of a higher performance of the corresponding PSCs, with enhanced photocurrent density and improved open circuit voltage values, owing to minimized recombination losses.

#### 2.2.3. Scanning Electron Microscopy (SEM)

The effect of the post-treatments on the morphology of the perovskite films has been elucidated by applying Scanning Electron Microscopy measurements in half cells of the FTO/c-TiO_2_/m-TiO_2_/ZrO_2_/perovskite structure, as depicted in Figure 11.

In the pictures of the reference (untreated) film, the appearance of PbI_2_ residuals on the surface is clear, which is consistent with the obtained XRD patterns, where a peak of unreacted PbI_2_ is present and is in accordance with the composition of the perovskite precursor solution, containing an excess of PbI_2_ [57,58]. When the films are post-treated with AI and EAI, the picture is similar: the traces of PbI_2_ on the surface of the film are minimized and even though pinholes continue to appear and the films are more homogeneous than the reference film. After post-treatment with PEAI and AVAI, the pinholes disappear and the films appear dense. A peculiar morphology is obtained after the TBAI treatment, where large crystals form on top of the perovskite film, completely covering the perovskite crystals. This is expected to be unfavorable for the device configuration under study, where the post-treatment agent needs to penetrate through the mesoporous C electrode.

#### 2.2.4. Solar Cells’ Characterization

In order to evaluate the performance of the perovskites in perovskite solar cells, the typical J-V curves were recorded under standard reporting conditions (air mass 1.5 (AM1.5) global solar light at 100 mW cm^−2^ and room temperature for all the fabricated devices (Figure 12)). The electrical parameters were extracted from the J-V curves and are summarized in Table 5.

Significantly improved performance of the C-PSCs, which have been prepared and subjected to post-treatment, has been achieved when ammonium iodide (AI) has been used as the post-treatment agent. A notable increase has been recorded in the J_sc_ values, reaching 22.1 mA/cm^2^ and the FF values, reaching 0.54, which has led to the highest PCE obtained of 9.77%, a 26% rise compared to 7.75% of the reference device. A similar effect takes place upon the application of ethylammonium iodide (EAI) as post-treatment agent. Current density values of 20.16 mA/cm^2^, which is higher than 19.8 mA/cm^2^ of the reference device, have been recorded and an FF of 0.54, which is by 14.9% enhanced compared to the reference of 0.47, while in the EAI case there is an additional improvement in the V_oc_ values by 10 mV, which raises the PCE to 9.16%. The increased photocurrent density values obtained for both AI and EAI post-treatment agents indicate a better crystal quality and a reduced trap density. Indeed, ammonium iodides have previously demonstrated an ability to promote perovskite enlargement and grain growth [59,60,61], something which appears to be universal and to be applied in all PSC device configurations, regardless of the electrode and structure. An improvement in the PCE has also been recorded after the post-treatment with 5–amino valeric acid (AVAI), which is entirely attributed to the significant improvement in the FF value, climbing up to 0.59. This is a clear indication of the improved perovskite/charge extraction layer interfaces. AVAI is well known to form a wide bandgap 2D perovskite (AVA_2_PbI_4_) capping layer, effectively passivating the surface defects and at the same time effectively suppressing the back electron transfer and reducing the current leakage losses [55,62]. On the other hand, despite the promising results obtained from the absorbance spectra, both tetrabutyl ammonium iodide (TBAI) and phenethyl ammonium iodide (PEAI) have yielded inferior PCEs compared to the reference device. This can be ascribed to their long alkyl chains, which inhibit their adequate infiltration through the mesoporous C electrode. By not properly penetrating the triple mesoscopic stack, the only effect of this post-treatment is the partial washing of the perovskite surface by the solvent used for the stock solution, in this case isopropanol, which leads to local partially deconstructed areas in the perovskite film and induces a drop in the J_sc_ values, which is mainly responsible for the drop in the PCE. This result highlights the major differences that the triple mesoscopic structured, C electrode PSCs present over the conventional structured, metal electrode PSCs, as well as the challenges regarding their fabrication process. It becomes clear that this is an entirely separate research area, which should not be compared to or conjugated with the conventional PSCs.

In order to investigate the effect of the post-treatment on the hysteresis of the C-PSCs, the hysteresis index (HI) of each device has been calculated using the formula $Hysteresis\;Index=\frac{PCE_{RS}-PCE_{FS}}{PCE_{RS}}$ (Table 6), while the J-V curves that have been recorded for the champion devices in forward and reverse scan mode are illustrated in Figure 13.

According to previous reports, ammonium iodides, particularly the ones with long alkyl chains such as PEAI (n = 2), are found to reduce the hysteresis due to improved carrier lifetimes [63]. However, this has not been observed in our system. In particular, all of the post-treated C-PSCs exhibit an increase in the HI, which is also found from the obtained J-V curves. In general, triple mesoscopic-structured C-PSCs suffer from an increased hysteresis due to the high thickness of each mesoporous layer, combined with the absence of Hole transport Layer (HTL) [64]. This phenomenon is attributed to the defects, bulk and interfacial, of the perovskite, which act as traps, the ferroelectric property of the perovskite and the interstitial defects causing ion migration [65]. Post-treatment of the perovskite surface has been proposed as a method to successfully alleviate hysteresis in the conventionally structured PSCs; however, the same has not appeared to apply for the devices under study, which is attributed to the vast differences in the structure, i.e., the porous C electrode, that the post-treatment solution must infiltrate through to reach the perovskite/electrode interface and at the complex structured perovskite that has been used, contrary to the most widely studied CH_3_NH_3_PbI_3_. The appearance of capacitive current, originating from ion migration and charge accumulation and creating capacitor components formed near the dual junctions of ETL/perovskite and perovskite/C, as proposed by Cojocaru et al. [66] can be an explanation for this behavior. Indeed, a high capacitance element has been recorded from Electrochemical Impedance Spectroscopy measurements at the intermediate frequency region, for the C-PSCs after their post-treatment, as will be discussed in Section 2.2.5, which can be correlated with electrode polarization caused by electric or ionic charge accumulation [67].

#### 2.2.5. Electrical Characterization

In order to further elucidate the charge transport and interface properties, Electrochemical Impedance Spectroscopy (EIS) has been applied in the C-PSCs, under illumination at an applied voltage of 0.8 V (close to the V_oc_). The Nyquist plots obtained are illustrated in Figure 14a and the Bode phase pots are illustrated in Figure 14b, while Table 7 summarizes the results obtained after the fitting of the spectra, by using an equivalent electrical circuit (Figure 7).

In this case, the two semicircles that appear in the Nyquist plot and correspondingly the peaks that appear at the Bode phase plot, are again distinct, as in the case of the ammonium salts used as additives. Similarly to the ammonium iodides that were used as additives, in the case of post-treatment, AI presents the lowest R_s_ value, in accordance with the improved FF values obtained for the corresponding C-PSCs. The same observation applies for the values of the charge transfer resistance (R_ct_), where the ammonium iodides of AI and EAI present the lowest values of 36.2 and 38.4 Ω respectively, and at the same time the corresponding devices exhibit the highest J_sc_ values. At the same time, all of the post-treatment methods achieve a diminishment in the R_ct_ values, a clear indication of the improved charge transport layer/perovskite interfaces that facilitates the efficient extraction of charges. The increased capacitive behaviour noted for the post-treated devices is an indication of capacitive currents produced after the post-treatments and correlated with the hysteresis of the C-PSCs, as reported in the previous section.

Even though the two compounds of AI and EAI also exhibit the lowest Zrec values, in accordance to the minor changes that the post-treatments provoke on the Voc values, the enhanced charge transport properties compensate for the potential losses, leading to an improved performance. However, it is very important to notice that the expected outcome of the post-treatments has been the opposite effect, an increase in the Zrec and the Voc values respectively. This has only been reported for the TBAI post treatment, which, however, presents an inferior performance overall, highlighting the major differences in the triple mesoscopic C-PSCs architecture, where the chemical interactions of the post-treatment agents with the mesoscopic C electrode defines the suitability and efficacy of any modification, contrary to the conventional PSCs that have been studied so far.

Contrary to the use of ammonium iodides as additives, when the same compounds are used as post-treatment agents the changes in the Bode phase plots are less profound. In particular, no frequency shifts have been observed, which indicates a similar electron lifetime for all C-PSCs under study, implying that no changes are induced in the bulk of the perovskite during the post-treatment and the only effects that this method has is on the surface of the perovskite film and the interfaces of the perovskite with the charge transport layers.

#### 2.2.6. Stability

In order to evaluate the stability of the as-prepared C-PSCs and the effect of each post-treatment agent, the highest performing devices have been stored in a dark shelf under ambient conditions (temperature ~35° C and humidity ~40%) for 60 days and the J-V measurements were repeated (Figure 15) and the electrical parameters have been obtained (Table 8).

Improved V_oc_ values have been recorded for all the C-PSCs, justifying that, in ambient fabrication conditions, the perovskite crystallization process continues even after the thermal annealing process [68] and it is something that our group has noted before in this particular device architecture [53]. All of the devices can be classified as stable, retaining over 80% of their initial PCE 60 days after their preparation. However, it is noteworthy that in certain cases, there is a significant increase in the electrical parameters and consequently the PCE, this being the post-treatment with TBAI and PEAI. These two compounds are the most hydrophobic among the ammonium iodides that have been studied, and therefore the corresponding devices are characterized by improved moisture stability [56]. The minimized degradation gives space for the defect-healing and re-crystallization process that has been reported to occur in C-PSCs [51,52]. Another explanation for this increase is the suppressed ion migration, which has also been reported in the literature [21].

## 3. Materials and Methods

The chemical formulas and structures of all ammonium iodide additives that have been used in this study are presented in Table 9.

Lead bromide (PbBr_2_) and methyl ammonium bromide (MABr) were purchased from TCI (Tokyo Chemical Industry). Formamidinium iodide was purchased from Dyenamo. All the other chemicals were purchased from Sigma-Aldrich. Commercial nanocrystalline Titania paste 18-NRT was purchased from Greatcell and SnO_2_: F transparent conductive electrodes (FTO, Resistance 10 Ohm/square) were purchased from Pilkington. The carbon and ZrO_2_ pastes were prepared according to the method described elsewhere [69].

The reference perovskite Cs_y_[(FAPbI_3_)_1−*x*_ (MAPbBr_3_)*_x_*]_1−y_ (*x* = 0.15 and y = 0.05) was prepared by mixing FAI (1.27 M), MABr (0.225 M), PbI_2_ (1.39 M) and PbBr_2_ (0.225 M) in DMF:DMSO (4:1 *v/v*) with subsequent addition of 5% *v/v* of CsI (1.5 M in DMSO). The solution was kept under stirring at 70 °C, for 2 h before deposition. The organic ammonium iodides used as additives (NH_4_I, EAI, TBAI, PEAI, 5-AVAI) were incorporated in the reference perovskite precursor solution by adding the proper amount of stock solution (0.8 M in DMF) in order to reach the desired final additive concentration (5 mol% with respect to FAI content).

Fluorine-doped tin oxide (FTO, 10 ohm/square) glass substrates were chemically etched by reacting Zn powder with 4 M HCl, to form the desired pattern. The etched substrates were subsequently immersed in surfactant solution (Triton-X 10% wt in distilled water), deionized water, acetone and isopropanol and placed in ultrasonic bath for 15 min per cleaning step. After the last rinse with isopropanol, the clean substrates were annealed at 500 °C, in order to remove any organic residuals. The compact TiO_2_ layer (c-TiO_2_) was deposited by spin coating 40 μL of Ti-diisopropoxide bis (acetyl) acetonate (75% in isopropanol, diluted by 1:9 *v/v* in *N*-propanol) at 2000 rpm for 10 s. The spin-coated films were dried at 125 °C and then annealed at 500 °C for 10 min and the procedure was repeated 4 times. The mesoporous TiO_2_ layer (m-TiO_2_) was then deposited, by spin-coating 40 μL of TiO_2_ paste (18-NRT, diluted in ethanol 1:6 *w/v*) at 4000 rpm for 20 s, followed by drying at 125 °C and annealing at 525 °C for 30 min. After cooling down, the ZrO_2_ was deposited on the substrates, by doctor blading a quantity of the as-prepared paste, followed by spin-coating at 5000 rpm for 30 s, in order to obtain a homogeneous film with full coverage. The films were then annealed at 450 °C for 20 min. Finally, the C counter electrodes were deposited by doctor-blading a quantity of the as-prepared paste, followed by annealing at 400 °C for 30 min. The perovskite solution was then infiltrated through the C electrode, followed by two-step spin coating at 1000 rpm for 10 s and 5000 rpm for 20 s. The perovskite-loaded substrates were then placed for annealing at 100 °C, for 60 min, capped with a petri dish to mitigate the impact of ambient oxygen and moisture [53]. No Hole Transport Material and no antisolvent dripping were used, while the whole process was performed entirely under ambient conditions (RT 20 °C and RH 25–30%). For the post-treatment of the perovskite films, after cooling down to room temperature, 80 μL of the ammonium iodide stock solutions (5 mg/mL in isopropanol) were dropped on the surface of the film, spin-coated at 5000 rpm for 20 s, and annealed at 100° for 10 min. A total of 8 C-PSCs for each scenario have been prepared and the champion devices’ performance is demonstrated.

The Ultraviolet-Visible/NIR absorption diffuse reflectance spectra of the perovskite films were obtained in a range of 300 nm to 850 nm using a Jasco V-770 spectrophotometer equipped with a 60 mm integrating sphere embedding a PbS Detector (ISN-923).

Photocurrent density-voltage (J-V) curves were recorded by using a Keithley 2601 source meter, with an applied potential range from 1.1 to −0.1 V and a scan rate of 125 mV/s, by illumination with a Solar Light simulator, providing a beam of 100 mWcm^−2^ light intensity. In order to determine the active area of the devices, a black metallic mask of aperture size of 0.145 cm^2^, which is very close to the actual area of the devices, was used [70]. External quantum efficiency (EQE) was determined by incident photo to current efficiency (IPCE) measurement by Thetametrisis PM−QE equipped with Xenon (Xe) light source using a filter monochromator (Oriel CornerstoneTM 260 1/4 m, Newport), which was controlled by PM−Monitor^®^ software. Electrochemical Impedance Spectroscopy (EIS) characterization was performed on PSC devices with a potentiostat/galvanostat (PGSTAT128 N, Autolab B.V., Netherlands) under AM-1.5G-illuminated conditions. The morphology was investigated with a high resolution field-emission scanning electron microscope (FE-SEM) instrument (Zeiss, SUPRA 35VP) operating at 10–15 kV. The crystal structure was investigated using X-ray Diffraction (XRD, Bruker D8 Advance) utilizing a nickel-filtered CuKα (1.5406 Å) radiation source operated at 40 kV and 40 mA with a fixed step size of 0.02° and 5 s/step in the range of 5° ≤ 2*θ* ≤ 60°.

## 4. Conclusions

In this work, the effect of ammonium iodides in PSCs with a C electrode and the triple mesoscopic structure has been evaluated. The study has been conducted in two directions: the incorporation of ammonium iodides as additives in the perovskite precursor solution and the application of ammonium iodides as post-treatment agents. In the direction of ammonium iodides as additives, after the incorporation of ammonium iodide (AI) in the precursor solution, an increase in the absorbance, combined with a decreased charge transfer resistance, has led to C-PSCs with PCE of 10.3%, increased by 33% compared to the reference perovskite, and with a low hysteresis index of 0.18, compared to 0.31 for the reference. In the direction of post-treatment with ammonium iodides, the two iodides with the shortest alkyl chain, ammonium iodide (AI) and ethylammonium iodide (EAI), have yielded the highest PCEs of 9.77% and 9.15% respectively, mainly attributed to the increased photocurrent density obtained, with a significant simultaneous increase in the FF values, which has confirmed the beneficial effect of the post-treatments in the optimizing of the perovskite/charge extraction layer interfaces. A common result of these parallel studies has been the influence of the alkyl chain length in the resulting properties of the C-PSCs. In particular, it has been noted in both cases that the longest alkyl chain is incapable of penetrating through the mesoporous C electrode, and therefore any favorable effect that the ammonium iodide could have had on the physical, chemical, structural and electrical properties is inhibited. The results of this work have demonstrated the importance of choosing the appropriate materials that will be compatible with each one of the individual parts that the C-PSCs comprise of, in order to obtain high performing devices. Finally, this study has revealed the key role that both the precursor solution and the post-treatment solution have when applying the ammonium iodides in the triple mesoscopic C electrode configuration, and has highlighted the great difference compared to the typical metal electrode PSCs, emphasizing the importance of independent research in this field.

## Figures and Tables

**Figure 1 molecules-26-05737-f001:**
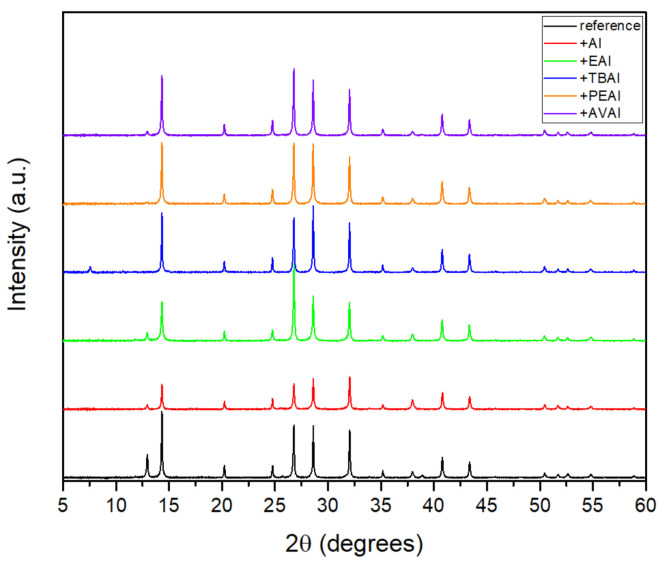
X-ray diffraction patterns of the perovskite films prepared without and with the ammonium iodide additives in the precursor solution.

**Figure 2 molecules-26-05737-f002:**
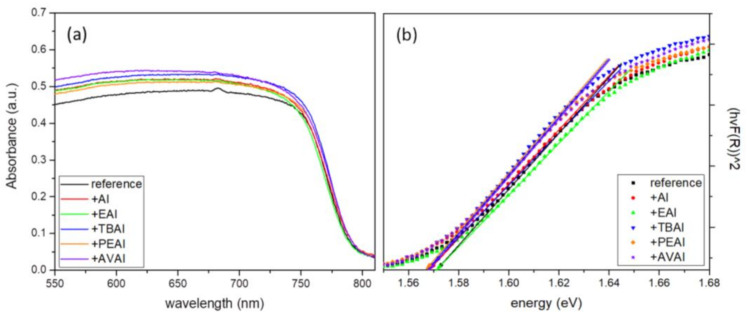
(**a**) UV-vis absorption spectra and (**b**) Tauc plots of the 6 perovskite films under study.

**Figure 3 molecules-26-05737-f003:**
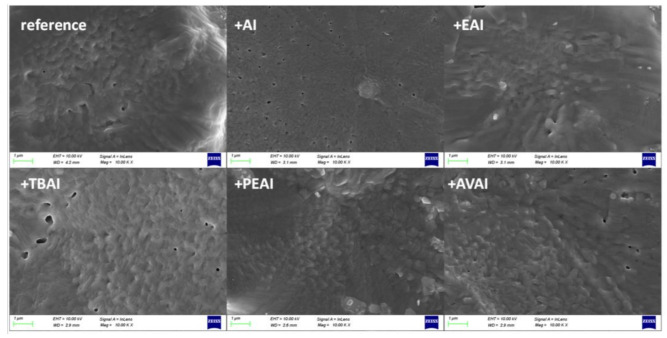
Top view SEM images obtained for the perovskite films with and without the ammonium iodides additives in the perovskite precursor solution.

**Figure 4 molecules-26-05737-f004:**
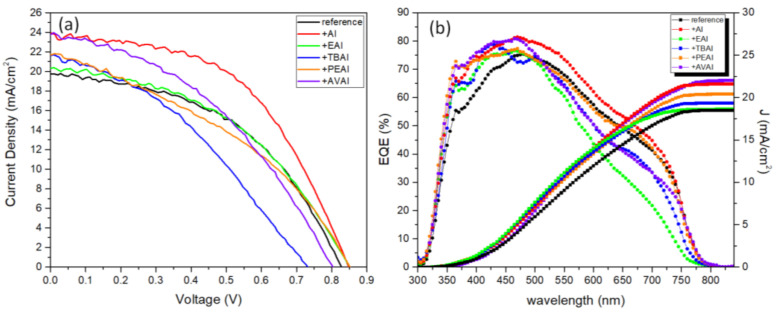
(**a**) Current density–voltage curves of carbon based solar cells, incorporating the 6 perovskites under study, under illumination of 100 mW cm^−2^ simulated sun irradiation (1.5 AM) and (**b**) EQE of the corresponding solar cells.

**Figure 5 molecules-26-05737-f005:**
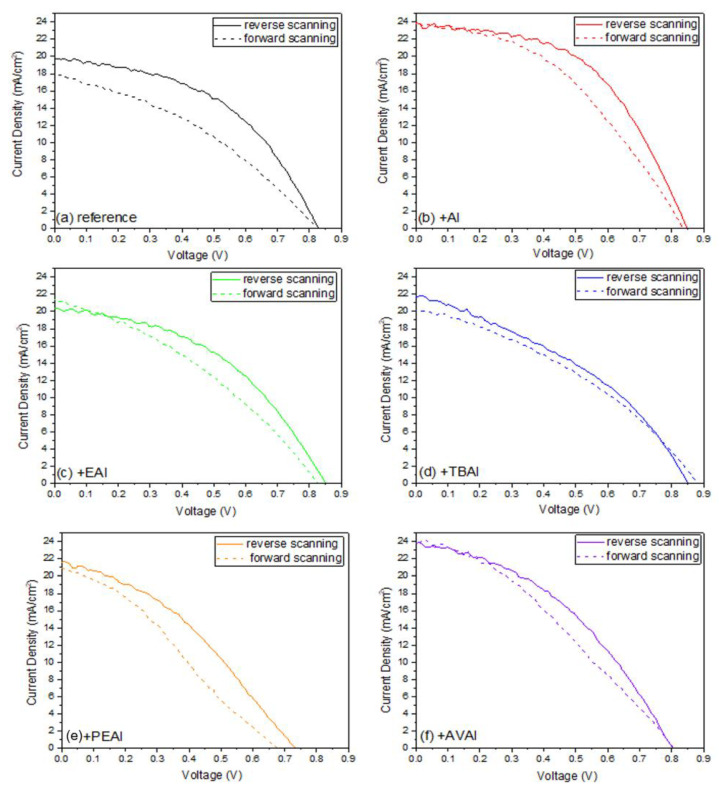
The J-V curves of the C-PSCs prepared with each perovskite under study (**a**) reference; (**b**) ammonium iodide additive; (**c**) ethyl ammonium iodide additive; (**d**) tetra butyl ammonium iodide additive; (**e**) phenethyl ammonium iodide additive; and (**f**) 5-ammonium valeric acid iodide.

**Figure 6 molecules-26-05737-f006:**
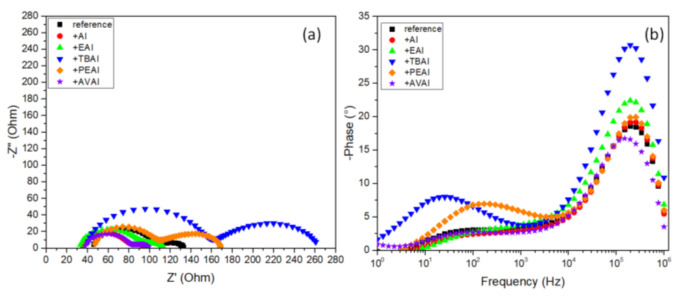
(**a**) Nyquist plots and (**b**) Bode phase plots obtained from Electrochemical Impedance Spectroscopy (EIS) measurements performed on the C-PSCs incorporating the 6 perovskites under study.

**Figure 7 molecules-26-05737-f007:**
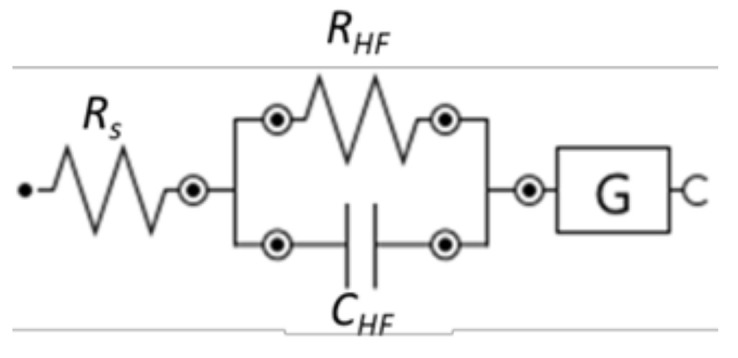
The equivalent electrical circuit that has been used to fit the data obtained from EIS measurements.

**Figure 8 molecules-26-05737-f008:**
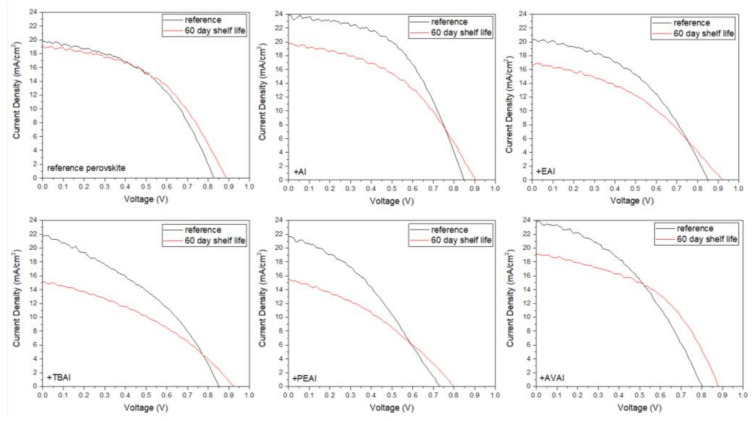
Comparative current density–voltage curves of the 6 carbon-based solar cells under study, under illumination of 100 mW cm^−2^ simulated sun irradiation (1.5 AM) at the first day of their fabrication and after 60 days of shelf life.

**Figure 9 molecules-26-05737-f009:**
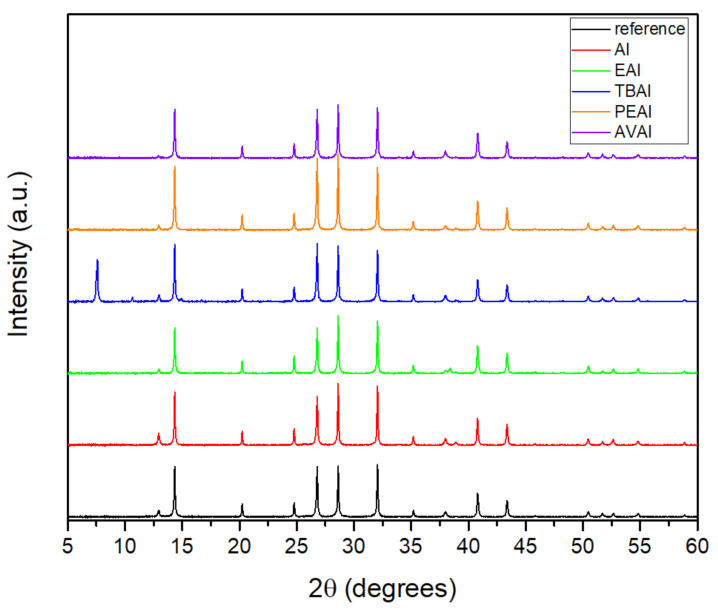
X-ray diffraction patterns of the perovskite films prepared without and with the ammonium iodide post-treatments.

**Figure 10 molecules-26-05737-f010:**
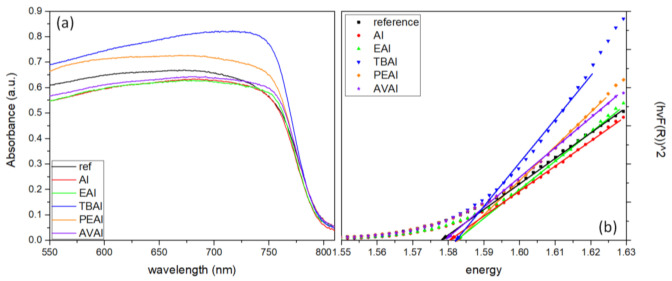
(**a**) UV-vis absorption spectra and (**b**) Tauc plots of the 6 perovskite films under study.

**Figure 11 molecules-26-05737-f011:**
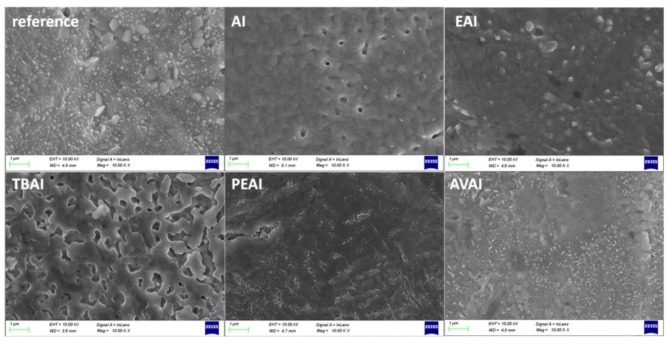
Top-view SEM images obtained for the perovskite films with and without post-treatment with the ammonium iodides under study.

**Figure 12 molecules-26-05737-f012:**
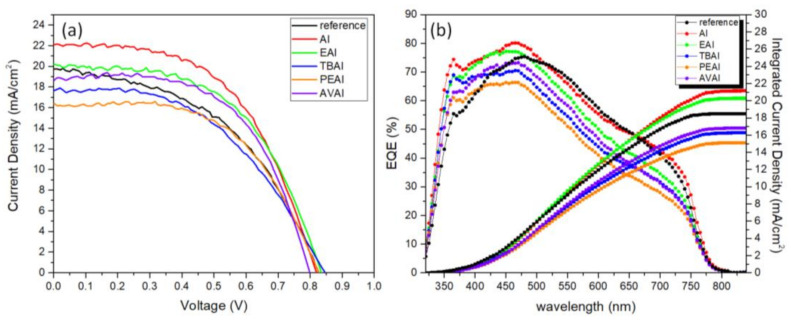
(**a**) Current density–voltage curves of the 6 carbon-based solar cells under study, under illumination of 100 mW cm^−2^ simulated sun irradiation (1.5 AM) and (**b**) EQE of the corresponding solar cells.

**Figure 13 molecules-26-05737-f013:**
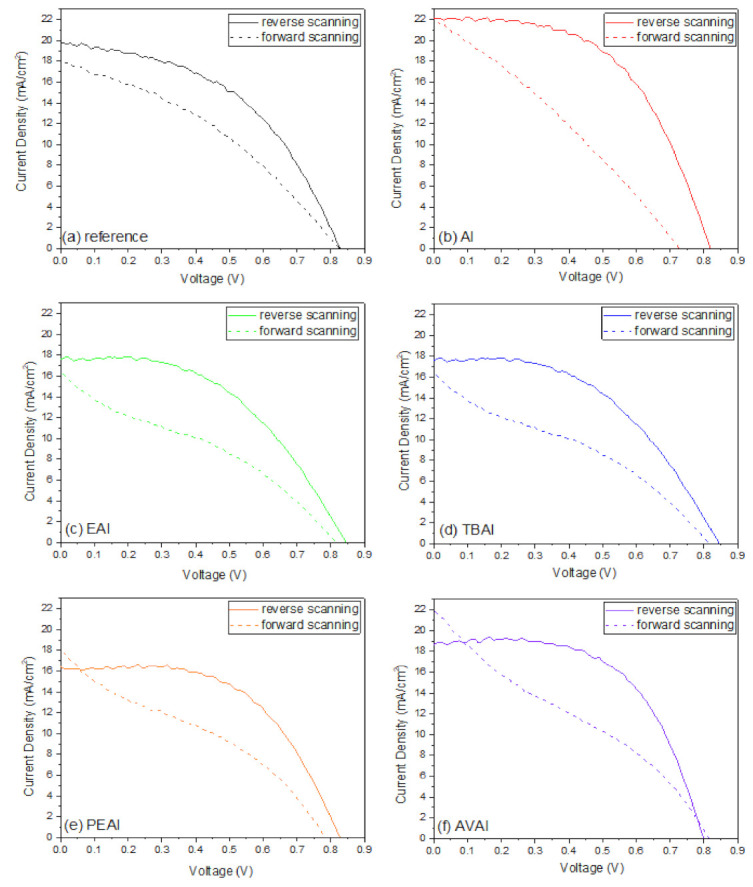
The J-V curves of the C-PSCs prepared with the reference and the post-treated devices, with the different post-treatment agents (**a**) reference; (**b**) ammonium iodide; (**c**) ethyl ammonium iodide; (**d**) tetra butyl ammonium iodide; (**e**) phenethyl ammonium iodide; and (**f**) 5-ammonium valeric acid iodide.

**Figure 14 molecules-26-05737-f014:**
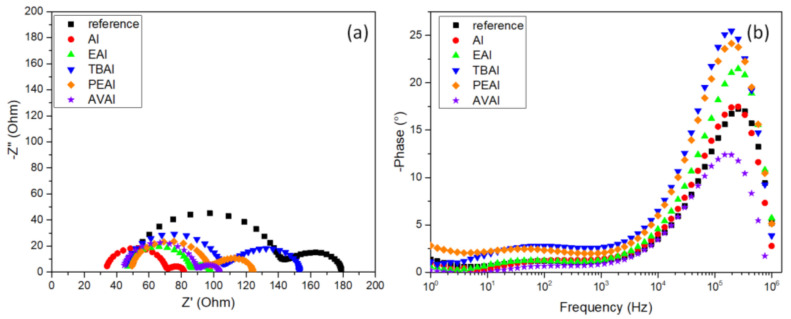
(**a**) Nyquist plots and (**b**) Bode phase plots obtained from Electrochemical Impedance Spectroscopy (EIS) measurements performed on the C-PSCs under study.

**Figure 15 molecules-26-05737-f015:**
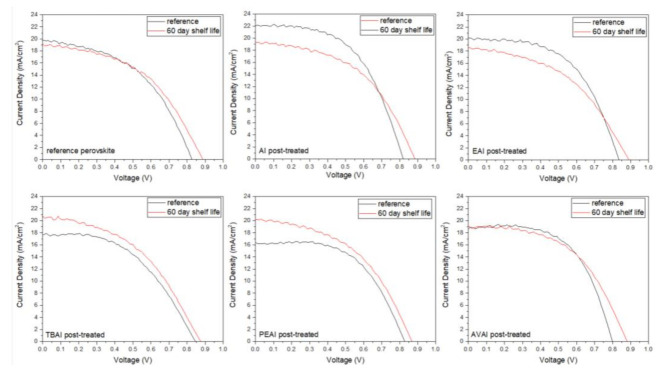
Comparative current density–voltage curves of the 6 carbon-based solar cells under study, under illumination of 100 mW cm^−2^ simulated sun irradiation (1.5 AM) at the first day of their fabrication and after 60 days of shelf life.

**Table 1 molecules-26-05737-t001:** Electrical parameters derived from the J-V curves of the champion C-PSCs, incorporating the reference (w/o additive) perovskite and the 5 additive-based perovskites under study.

Perovskite	J_sc_ (mA/cm^2^)	V_oc_ (V)	FF	PCE (%)	J_int._ (mA/cm^2^)
Reference	19.8	0.832	0.47	7.75	18.50
+AI	23.9	0.852	0.51	10.3	21.59
+EAI	20.3	0.852	0.45	7.74	18.64
+TBAI	21.7	0.852	0.38	7.00	19.37
+PEAI	21.7	0.733	0.36	5.73	20.41
+AVAI	24.0	0.803	0.41	7.77	22.04

**Table 2 molecules-26-05737-t002:** Summary table of the Hysteresis Index (HI) that has been calculated for each one of the C-PSCs under study.

Perovskite	HI
Reference	0.31
+AI	0.18
+EAI	0.20
+TBAI	0.08
+PEAI	0.24
+AVAI	0.16

**Table 3 molecules-26-05737-t003:** The electrical parameters obtained after fitting of the experimental data of the as-prepared devices’ Nyquist plots.

Perovskite	R_s_ (Ohm)	R_ct_ (Ohm)	C_ct_ (nf)	Y_0_ (mMho)	Z_rec_ (Ohm)
Reference	44.97	62.14	30.8	0.96	1.04
+AI	34.19	47.27	25.8	1.57	0.64
+EAI	32.34	58.44	29.1	1.08	0.93
+TBAI	39.04	121.28	17.2	1.11	0.91
+PEAI	46.51	62.45	22.9	0.73	1.37
+AVAI	37.57	44.98	42.4	2.19	0.46

**Table 4 molecules-26-05737-t004:** Comparative table of electrical parameters derived from the J-V curves of the C-PSCs under study at the first day of their fabrication and after 60 days of shelf life.

Perovskite	Day	J_sc_ (mA/cm^2^)	V_oc_ (V)	FF	PCE (%)	ΔPCE (%)
Reference	1	19.8	0.832	0.47	7.75	+3.61
	60	19.1	0.889	0.47	8.03	
+AI	1	23.9	0.852	0.51	10.3	−21.84
	60	19.8	0.910	0.45	8.05	
+EAI	1	20.3	0.852	0.45	7.74	−19.37
	60	16.9	0.925	0.40	6.24	
+TBAI	1	21.7	0.852	0.38	7.00	−26.00
	60	15.1	0.923	0.37	5.18	
+PEAI	1	21.7	0.733	0.36	5.73	−25.51
	60	15.4	0.803	0.36	4.36	
+AVAI	1	24.0	0.803	0.41	7.77	+2.19
	60	19.1	0.880	0.47	7.94	

**Table 5 molecules-26-05737-t005:** Electrical parameters derived from the J-V curves of the champion C-PSCs under study.

Perovskite	J_sc_ (mA/cm^2^)	V_oc_ (V)	FF	PCE (%)	J_int._ (mA/cm^2^)
Reference	19.8	0.832	0.47	7.75	18.50
AI	22.1	0.824	0.54	9.77	21.17
EAI	20.16	0.842	0.54	9.16	20.30
TBAI	17.7	0.852	0.48	7.3	16.29
PEAI	16.4	0.834	0.56	7.62	15.14
AVAI	18.6	0.805	0.59	8.85	16.83

**Table 6 molecules-26-05737-t006:** Summary table of the Hysteresis Index (HI) that has been calculated for each one of the C-PSCs under study.

Perovskite	HI
Reference	0.31
AI	0.51
EAI	0.63
TBAI	0.41
PEAI	0.39
AVAI	0.41

**Table 7 molecules-26-05737-t007:** The electrical parameters obtained after fitting of the experimental data of the as-prepared devices’ Nyquist plots.

Perovskite	R_s_ (Ohm)	R_ct_ (Ohm)	C_ct_ (nF)	Y_0_ (mMho)	Z_rec_ (Ohm)
Reference	45.56	96.85	19.1	0.96	1.04
AI	33.42	38.39	34.6	2.51	0.40
EAI	46.42	39.63	35.4	2.23	0.45
TBAI	45.91	60.34	27.0	0.92	1.10
PEAI	50.16	47.60	35.3	1.32	0.76
AVAI	44.32	45.19	30.9	2.17	0.46

**Table 8 molecules-26-05737-t008:** Comparative table of electrical parameters derived from the J-V curves of the C-PSCs under study at the first day of their fabrication and after 60 days of shelf life.

Perovskite	Day	J_sc_ (mA/cm^2^)	V_oc_ (V)	FF	PCE (%)	ΔPCE (%)
Reference	1	19.8	0.832	0.47	7.75	+3.61
	60	19.1	0.889	0.47	8.03	
AI	1	22.1	0.824	0.54	9.77	−13.51
	60	19.3	0.884	0.49	8.45	
EAI	1	20.16	0.842	0.54	9.16	−17.14
	60	18.6	0.899	0.45	7.59	
TBAI	1	17.7	0.852	0.48	7.3	+13.70
	60	20.6	0.880	0.45	8.3	
PEAI	1	16.4	0.834	0.56	7.62	+8.92
	60	20.2	0.871	0.47	8.3	
AVAI	1	18.6	0.805	0.59	8.85	−2.03
	60	19.1	0.889	0.51	8.67	

**Table 9 molecules-26-05737-t009:** Chemical formulas and structures of the ammonium iodides.

Compound	Chemical Formula	Chemical Structure
Ammonium Iodide (AI)	NH_4_I	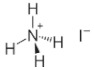
Ethyl ammonium Iodide (EAI)	CH_3_CH_2_NH_3_I	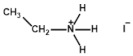
Tetra butyl ammonium Iodide (TBAI)	[CH_3_(CH_2_)_3_]_4_NΙ	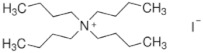
Phenethyl ammonium Iodide (PEAI)	(C_6_H_6_)(CH_2_)_2_NH_3_I	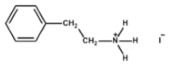
5-ammonium valeric acid Iodide (AVAI)	HO(O)C(CH_2_)_4_NH_3_I	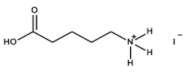

## Data Availability

Data is contained within the article.

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
