# Peer review of "Exploring the Effect of Ammonium Iodide Salts Employed in Multication Perovskite Solar Cells with a Carbon Electrode"

_molecules, 2021, doi:10.3390/molecules26195737_

Round 1
Reviewer 1 Report
The authors demonstrated the combined additive engineering and surface passivation using a series of ammonium iodides into the perovskite films, enabling the improved performance of carbon-electrode-based PSCs with suppressed non-radiative recombination. This manuscript is well-organized, and the experimental work is intensive and systematic. Thus, I recommend this manuscript be published on Molecules after major revision. To further bring out the significance of this work, the authors should address the following suggestion/comments below:
- The authors claimed that only TBAI can form the 2D perovskite structure in the bulk, whereas the others cannot. The authors should give some comments on the unique property of TBAI to form 2D structure.
- The authors demonstrated the suppressed non-radiative recombination after additive engineering and surface passivation. While, from the J-V curves, the obvious improvement is Jsc, not Voc. I don’t think the incorporation of ammonium iodides into the perovskite can enhance the absorption significantly if the perovskite thickness is similar.
- After post-treatment of surface passivation. Can these ammonium iodides form 2D perovskite structure at the top surface of the active layer? grazing incident XRD is recommended to investigate the possible gradient structure (from quasi-2D at the surface to 3D in the bulk).
- Did these ammonium iodides improve the long-term stability of both perovskite films and devices? Different soaking condition (e.g., light illumination, high humidity, thermal stress) is recommended to monitor the shelf and operational stability of C-PSCs.
Author Response
The authors demonstrated the combined additive engineering and surface passivation using a series of ammonium iodides into the perovskite films, enabling the improved performance of carbon-electrode-based PSCs with suppressed non-radiative recombination. This manuscript is well-organized, and the experimental work is intensive and systematic. Thus, I recommend this manuscript be published on Molecules after major revision. To further bring out the significance of this work, the authors should address the following suggestion/comments below:
Comment 1: The authors claimed that only TBAI can form the 2D perovskite structure in the bulk, whereas the others cannot. The authors should give some comments on the unique property of TBAI to form 2D structure.
Reply: We thank the reviewer for the constructive point. A comment is inserted in line 137, as suggested by the reviewer regarding the TBAI ability to form a 2D structure. In particular, we added the sentence:
“It is contemplated that the TBA cation could be pushed vertically towards the surface of the perovskite grains due to its large size and high level of hydrophobicity, compared to the other additives under study, being therefore more likely to form a 2D layered phase in the perovskite / C interface”
Comment 2: The authors demonstrated the suppressed non-radiative recombination after additive engineering and surface passivation. While, from the J-V curves, the obvious improvement is Jsc, not Voc. I don’t think the incorporation of ammonium iodides into the perovskite can enhance the absorption significantly if the perovskite thickness is similar.
Reply: We thank the reviewer for his/her constructive comment. We agree with that and it has been confirmed from the UV-vis measurements that only a small increase in the absorption occurs after the incorporation of the additives. The increase that has been observed in the corresponding C-PSCs has been attributed to a better crystal quality (line 202) and a better pore filling ability which leads to better coverage, in the triple mesoscopic structure under study.
Comment 3: After post-treatment of surface passivation. Can these ammonium iodides form 2D perovskite structure at the top surface of the active layer? grazing incident XRD is recommended to investigate the possible gradient structure (from quasi-2D at the surface to 3D in the bulk).
Reply: Unfortunately there is no capability for this measurement in our facilities and nearby. However we do not believe that the ammonium iodides, except for TBAI, can form 2D structures, considering that the post treatment was not followed by high temperature annealing, as also suggested in literature (Jiang, Q., Zhao, Y., Zhang, X. et al. Surface passivation of perovskite film for efficient solar cells. Nat. Photonics 13, 460–466 (2019). https://doi.org/10.1038/s41566-019-0398-2).
Comment 4: Did these ammonium iodides improve the long-term stability of both perovskite films and devices? Different soaking condition (e.g., light illumination, high humidity, thermal stress) is recommended to monitor the shelf and operational stability of C-PSCs.
Reply: The shelf life stability of the devices has been evaluated, by recording the J-V curves of the highest performing C-PSCs 60 days after their preparation and storage in ambient conditions (temperature ~35° C & humidity ~40%) and the results have been added in the manuscript in two additional Sections (Section 2.1.6 line 351 and 2.2.6 line 596), as proposed by the reviewer.
Reviewer 2 Report
In this manuscript entitled “Exploring the effect of ammonium iodide salts employed in multication perovskite solar cells with a Carbon electrode” authors report enhanced PCEs of carbon-based perovskite solar cells through a series of ammonium iodides as additives in perovskite precursor. The manuscript is not novel enough and the figure quality is poor. However, I believe the outputs could give the research community positive impact. However, it is hard to recommend publication of this manuscript in Molecules with current version, the author’s interpretation and description in this manuscript including a number of points that would be inappropriate and unclear.
- The y-axis (R%) of UV-vis absorption spectra is totally wrong, the authors should make a correction.
- The authors utilize the additives to post-treatment, generally, the surface defects should be passivated or eliminated resulting in an increased Voc. However, we did observe a decreased in Voc. Why?
- Could the authors give more reasons regarding the significant drop in the long wavelength (600nm-800nm) of EQE (Figure 4b and Figure 11b)?
- In the manuscript, the authors provide a relative weak evidence and description, just based on absorption, to explain the interaction between additives and perovskite, it would be better if the authors discuss more.
Author Response
In this manuscript entitled “Exploring the effect of ammonium iodide salts employed in multication perovskite solar cells with a Carbon electrode” authors report enhanced PCEs of carbon-based perovskite solar cells through a series of ammonium iodides as additives in perovskite precursor. The manuscript is not novel enough and the figure quality is poor. However, I believe the outputs could give the research community positive impact. However, it is hard to recommend publication of this manuscript in Molecules with current version, the author’s interpretation and description in this manuscript including a number of points that would be inappropriate and unclear.
Comment 1: The y-axis (R%) of UV-vis absorption spectra is totally wrong, the authors should make a correction.
Reply: We thank the reviewer for the careful consideration of our manuscript. The y-axis has been properly corrected according the reviewer’s comment and converted to Absorbance Units as suggested (Figure 4 and Figure 10).
Comment 2: The authors utilize the additives to post-treatment, generally, the surface defects should be passivated or eliminated resulting in an increased Voc. However, we did observe a decreased in Voc. Why?
Reply: We thank the reviewer for his/her constructive comment. In the device architecture under study, which is the triple mesoscopic, where both the perovskite precursor solution and the post-treatment solutions are inserted through infiltration of the mesoporous C electrode, a crucial factor that determines the device performance, as well as any effects that take place after the crystallization of the perovskite (such as any post-treatment method), is the ability of the solution to penetrate through the pores of the C electrode. This is determined by several factors, including the physical properties (e.g. solvent viscosity) and the chemical properties (e.g. hydrophilicity) of the solution. This is very profound in the cases of TBAI and PEAI post-treatment agents. The reason for the absence of Voc enhancement in our study is that the perovskite crystals are “confined” into the pores of the triple mesoscopic layer (TiO2/ZrO2/C), and particularly the C layer. In this way the defects that in other PSC structures appear on the perovskite film surface are basically defects at the grain boundaries and at the perovskite/ZrO2 – perovskite/C interfaces, which are not affected by the post-treatment. However, there is an improvement of these interfaces after the post-treatment, which has been confirmed by EIS measurements and is responsible for the FF improvement which has led to higher PCEs. Overall, the reason for the declination of the obtained results from the typically reported behavior lies in the major difference and uniqueness of the triple mesoscopic PSC structure, which is highlighted throughout the manuscript and has been the motivation behind this work.
Comment 3: Could the authors give more reasons regarding the significant drop in the long wavelength (600nm-800nm) of EQE (Figure 4b and Figure 11b)?
Reply: This form of the EQE spectra obtained has been previously demonstrated in literature and is related to both the triple mesoscopic structure under study (TiO2/ZrO2/C) and the Hole Transport Layer. In particular, the significant drop after 600 nm is commonly reported in HTL-free devices, like the system under study (e.g. 1. Xu, L.,. et al. Nanoscale Res Lett 12, 159 (2017), 2. Lv, M. et.al. ACS Appl. Mater. Interfaces 2015, 7, 31, 17482–17488, 3. Zhou, L., Zuo, Y., Mallick, T.K. et al. Sci Rep 9, 8778 (2019), 4. Zhang, H., Lv, Y., Guo, Y. et al. J Mater Sci: Mater Electron 29, 3759–3766 (2018).)
Comment 4: In the manuscript, the authors provide a relative weak evidence and description, just based on absorption, to explain the interaction between additives and perovskite, it would be better if the authors discuss more.
Reply: The following comment has been added in the text (line 202) together with additional references.
“The varied cations (A+, EA+, TBA+, PEA+ and AVA+) intercalate between MA+/FA+ and [PbI6]4- through diverse intermolecular interactions (mainly hydrogen bonds), affecting the quality of perovskite films. The strength of this intermolecular hydrogen bonding is expected to have an effect on the crystallization kinetics, slowing down the crystal growth rate during the thermal annealing, and delivering a more ordered and homogeneous perovskite film, with fewer unordered aggregation of perovskite particles on the mesoporous films [19,41].”
An in-depth discussion can also be found in Section 2.1.4, lines 226-249.
Reviewer 3 Report
In this paper, the authors present the effect of different ammonium salts in the perovskite precursor solution as an additive in Carbon-based Perovskite Solar Cells.
However, I have the following comments.
My concerns are the IV curves and EQE curves. As you can see from EQE curves, maximum around 70% and 60% from Figure 4b and Figure 11 b. The calculation of Integrated Jsc was wrong for both Figures. If you consider the EQE with 70% for the 350 to 800 nm whole range, you will only get the Jsc around 19mA/cm2. However, the authors reported Integrated Jsc values are unrealistic. It contains factual errors.
It clearly shows that all the IV curves are wrong and overestimating the photocurrent.
There were some problems with IV measurements.
I do not recommend this manuscript for publication.
Author Response
Comment 1: In this paper, the authors present the effect of different ammonium salts in the perovskite precursor solution as an additive in Carbon-based Perovskite Solar Cells. However, I have the following comments. My concerns are the IV curves and EQE curves. As you can see from EQE curves, maximum around 70% and 60% from Figure 4b and Figure 11 b. The calculation of Integrated Jsc was wrong for both Figures. If you consider the EQE with 70% for the 350 to 800 nm whole range, you will only get the Jsc around 19mA/cm2. However, the authors reported Integrated Jsc values are unrealistic. It contains factual errors. It clearly shows that all the IV curves are wrong and overestimating the photocurrent. There were some problems with IV measurements. I do not recommend this manuscript for publication.
Reply: We thank the reviewer for carefully reading our paper. Motivated by this meaningful comment we re-evaluated the parameters of the software that we have used for our measurements and have found a systematic error in the EQE data manipulation and we apologize for that. This was corrected and the EQE measurements were repeated for our devices (Figure 4 & Figure 12). The updated data have been inserted in the Tables 1 & 5. The new results have also been cross-checked to be in accordance to the methods proposed in literature for the reliable solar cell measurements, (e.g. 1. Michael Saliba and Lioz Etgar, ACS Energy Letters 2020 5 (9), 2886-2888, 2. Christians, Jeffrey A., Manser, Joseph S., Kamat, Prashant V. J. Phys. Chem. Lett. 2015, 6, 5, 852–857)
Reviewer 4 Report
This is an interesting study that introduces several new approaches. Could you please in the introduction explain a little about the difference between the use of the salts as additives and as passivating agents? Some citations should be added and the purpose of the two approaches should be better explained and justified.
line 97 You are changing cations not anions, the iodine remains the same, right?
Line 118 How does the disappearance of unreacted PbI2 on insertion of additives tell you anything about the suppression of the formation of inactive species? Please explain better, this makes no sense.
Fig 5 Please put all graphs on the same scale. That would be much better for making the comparison.
Author Response
This is an interesting study that introduces several new approaches.
Comment 1: Could you please in the introduction explain a little about the difference between the use of the salts as additives and as passivating agents? Some citations should be added and the purpose of the two approaches should be better explained and justified.
Reply: The following paragraph has been added in the introduction (line 81) of the manuscript, along with the corresponding references:
“By introducing the ammonium iodides as additives in the perovskite precursor solution, chemical interactions between the additive and the perovskite occur, with the most prominent being the hydrogen bonding between the inorganic framework and the ammonium group. This interaction is responsible for the ordering and packing of the molecular domains, while it additionally determines the geometry and properties of the perovskite crystal, by regulating the crystal growth[28,29].
On the other hand, the post-treatment of the perovskite film, in the annealed perovskite phase, using the ammonium iodides as post-treatment agents is expected to reduce the trap density at the surface of the perovskite layer and consequently improve the performance and stability[30,31]. By using this method, a passivation occurs at the perovskite surface, which makes the material less prone to external environment effects and a suppress of the ion movement, together with reduction of defects is achieved[32].”
Comment 2: Line 97 You are changing cations not anions, the iodine remains the same, right?
Reply: Yes, the anion is iodine in every case was studied.
Comment 3: Line 118 How does the disappearance of unreacted PbI2 on insertion of additives tell you anything about the suppression of the formation of inactive species? Please explain better, this makes no sense.
Reply: By using the term “inactive species” we refer to the unreacted PbI2 fragments, that only appear in the reference perovskite, while for the additive-based samples the corresponding XRD peak is absent. This is an indication that the additives favor the complete conversion of the perovskite precursor to perovskite crystals and suppress the formation of PbI2.
Comment 4: Fig 5 Please put all graphs on the same scale. That would be much better for making the comparison.
Reply: The scale of the graphs in both Fig.5 and Fig. 13 (former Fig.12) has been changed as proposed by the reviewer.
Round 2
Reviewer 1 Report
The authors try to address the comments, however, the reviewer still believes that certain points warrant more attention. For example, the integrated Jsc from EQE is apparently wrong. The authors should double-check the EQE spectra and intergrated values.
Author Response
we kindly provide our response to the attached file

Reviewer 3 Report
The authors have addressed the comments from the reviewer. I think the revised manuscript can be accepted for publication
Author Response
We thank the reviewer for his/her positive opinion and we are glad that we satisfied him/her with our replies